# Preanalytical considerations for clinical assays of circulating human miRNA-451a, miRNA-423-5p and miRNA-199a-3p for diagnostic purposes

**Dinesh S. Chandel, Wesley A. Tom, Chao Jiang, Gary Krzyzanowski, Nirmalee Fernando, Appolinaire Olou, M. Rohan Fernando** *

Molecular Diagnostic Research Laboratory, Center for Sensory Neuroscience, Boys Town National Research Hospital, Omaha, NE, United States of America

* M.Rohan.Fernando@boystown.org

## Abstract

Circulating miRNA has recently emerged as important biomolecules with potential clinical values as diagnostic markers for several diseases. However, to be used as such, it is critical to accurately quantify miRNAs in the clinic. Yet, preanalytical factors that can affect an error-free quantification of these miRNAs have not been explored. This study aimed at investigating several of these preanalytical factors that may affect the accurate quantification of miRNA-451a, miRNA-423-5p and miRNA-199a-3p in human blood samples. We initially evaluated levels of these three miRNAs in red blood cells (RBCs), white blood cells (WBCs), platelets, and plasma by droplet digital PCR (ddPCR). Next, we monitored miRNA levels in whole blood or platelet rich plasma (PRP) stored at different temperatures for different time periods by ddPCR. We also investigated the effects of hemolysis on miRNA concentrations in platelet-free plasma (PFP). Our results demonstrate that more than 97% of miRNA-451a and miRNA-423-5p in the blood are localized in RBCs, with only trace amounts present in WBCs, platelets, and plasma. Highest amount of the miRNA-199a-3p is present in platelets. Hemolysis had a significant impact on both miRNA-451a and miRNA-423-5p concentrations in plasma, however miRNA-199a levels remain unaffected. Importantly, PRP stored at room temperature (RT) or 4˚C showed a statistically significant decrease in miRNA-451a levels, while the other two miRNAs were increased, at days 1, 2, 3 and 7. PFP at RT caused statistically significant steady decline in miRNA-451a and miRNA-423-5p, observed at 12, 24, 36, 48 and 72 hours. Levels of the miRNA-199a-3p in PFP was stable during first 72 hours at RT. PFP stored at -20˚C for 7 days showed declining stability of miRNA-451a over time. However, at -80˚C miRNA-451a levels were stable up to 7 days. Together, our data indicate that hemolysis and blood storage at RT, 4˚C and -20˚C may have significant negative effects on the accuracy of circulating miRNA-451a and miRNA-423-5p quantification.

**Data Availability Statement:** All relevant data are within the manuscript.

**Funding:** This research was funded by a research grant from Ryan Foundation given to MRF. The funders had no role in study design, data collection and analysis, decision to publish, or preparation of the manuscript.

**Competing interests:** The authors declare no conflict of interest.

## Introduction

In clinical chemistry, the development of minimally invasive diagnostic and screening assays based on cell-free nucleic acids has garnered increasing attention. Cell-free DNA, mRNA, and microRNAs (miRNAs) found in blood have emerged as prominent biomarkers in the pursuit of assay innovation. However, a critical challenge inherent to the utilization of blood-based biomarkers is the influence of pre-analytical factors, such as the way the blood sample is handled, including the storage and processing, which might exert pronounced impacts on assay outcomes [1].

MiRNAs are a class of small non-coding RNA molecules, typically comprising around 22 nucleotides, with pivotal roles in pathophysiological processes. They function by post-transcriptionally regulating gene expression mostly through gene silencing [2, 3]. MiRNAs play significant roles in several cellular processes including differentiation, proliferation, apoptosis, and tumorigenesis [4]. Recent research has unveiled the dysregulation of miRNA expression in various disease conditions, including cancer, chronic kidney disease, and autism spectrum disorder [5–7]. Notably, investigations carried out on fresh human tissues and formalin-fixed paraffin-embedded (FFPE) tissues have underscored the remarkable stability of miRNAs [8, 9]. Likewise, studies involving human plasma have indicated the relative stability of plasma miRNAs, bolstering their potential as diagnostic biomarkers [10]. However, a 2022 study revealed the degradation of certain miRNAs after 24 hours of storage at room temperature [11].

Among these miRNAs, as stated above, miRNA-451a, miRNA-423-5p and miRNA-199a-3p emerged as key player in various human diseases [12–17]. MiRNA-451a has been implicated in a spectrum of cancers, such as esophageal cancer [18], gastric cancer [19], renal cell carcinoma [20], hepatocellular carcinoma [21], colorectal cancer [22], and breast cancer [23]. As such, miRNA-451a was recently proposed as a potential biomarker and therapeutic target across multiple cancer types [24]. MiRNA-423-5p is known for its high abundance in plasma and has been reported as an endogenous control for the quantification of circulating miRNAs in certain cancers [16]. MiRNA-423-5p has been implicated in somatotroph adenomas [25], colorectal carcinoma [26], ovarian cancer [27] and brain metastasis [28]. MiRNA-199a-3p is another miRNA that is highly abundant in blood and is a promising diagnostic and prognostic marker in glioma [17]. MiRNA-199a-3p has also been implicated in gastric cancer [29] and diabetic neuropathy [30].

Considering that most errors in clinical laboratory tests occur during the pre-analytical phase [31], diligent monitoring and control of pre-analytical factors are paramount to minimizing inaccuracies in clinical assays. Sample type, storage temperature, and time of sample processing can all alter, to varying degree, the expression of miRNAs. Thus, if miRNAs are going to be used in diagnostic assays in the future, it is critical to characterize these pre-analytical factors to generate consistent and accurate conclusions about miRNA expression levels. Given the extensive interest in miRNA-451a, miRNA-423-5p and miRNA-199a-3p as putative biomarkers for disease, this study sought to investigate pre-analytical factors that may exert significant effects on the accurate quantification of these plasma miRNAs in blood. Here, we provide a set of empirical guidelines which can help minimize misinterpretation of miRNA dysregulation in blood samples. Comparing three distinct miRNAs, we demonstrate that pre-analytical conditions may differentially impact the stability of many disease-specific miRNAs, and therefore diagnostic assays based on individual miRNA dysregulation be interpreted with extra caution.

## Methods and materials

### Blood samples

Blood from healthy adult donors were obtained from Boys Town National Research Hospital (BTNRH), Omaha, NE, USA, during the period between October 2020 and December 2023. Written informed consent was obtained from all donors prior to blood draw and this study was approved (IRB # 20-14-XP) by the institutional review board at BTNRH. Blood was collected from each donor using standard venipuncture technique into one 10 mL $K_3$EDTA tube (BD vacu-tainer1, Becton Dickinson, Franklin Lakes, NJ). To study the individual miRNA concentrations in different fractions of human blood, we used freshly drawn blood from four healthy donors. Blood from seventeen donors were used to study the effect of storage of whole blood and PRP at RT or 4°C for up to 7-days on stability of the circulating miRNA-451a. Blood from eight donors were used in the 7-day stability study for miRNA-423-5p and miRNA-199a-3p in whole blood and PRP at RT or 4°C. For short term study conducted at RT for 72 hours and low temperature study conducted at -20°C and -80°C, blood was drawn from six donors.

### Plasma separation

Platelet rich plasma (PRP) was prepared from blood by a one-step low speed centrifuging blood samples at room temperature (RT) at 1600 x g for 10 minutes, plasma layer was then carefully removed without disturbing the buffy coat and transferred into a new tube. Platelet free plasma (PFP) was prepared from whole blood by a previously described method [32]. Blood was centrifuged at room temperature (RT) at 1600 x g for 10 minutes, plasma layer was then carefully removed without disturbing the buffy coat, transferred into a new tube for a second spin at high speed, centrifuged at RT at 16000 x g for 10 minutes, to remove platelets, residual cells, cell debris, apoptotic bodies and nuclei.

### Preparation of artificially hemolyzed blood samples

Immediately after blood draw, blood was divided into two aliquots. One aliquot was subjected to freezing and thawing to induce hemolysis. The other aliquot was used as non-hemolyzed control sample. Plasma was separated from both hemolyzed and non-hemolyzed blood using two step centrifugation protocol as described above. RNA was extracted from both non-hemolyzed and hemolyzed plasma samples as described below.

### Plasma RNA extraction

Manufacturer's recommended protocol was followed to extract RNA from plasma using miR-Neasy® Serum / Plasma kit (Cat. No. / ID: 217184 QIAGEN Sciences Inc., Germantown, MD). RNA was eluted in 30 μL of nuclease free water. Total RNA concentration was determined using Qubit™ RNA HS Assay Kit (Life Technologies, Thermo Fisher Scientific Inc.), Qubit™ 4.0 Fluorometer (Life Technologies, Thermo Fisher Scientific Inc.) and stored at -80°C until cDNA transcription.

### Quantitative analysis of miRNAs using ddPCR

Total RNA (10 ng) was transcribed and amplified using TaqMan Advanced cDNA Synthesis Kit (Cat. # A28007, Applied Biosystems) following manufacturer's recommended protocol. Resulting cDNA samples were diluted with 0.1X TE buffer prior to individual miRNA probe specific ddPCR analysis using TaqMan advanced miRNA assays for miRNA-451a (Assay ID 478107-mir), miRNA-423-5p (Assay ID 478090-mir), and miRNA-199a-3p (Assay ID 477961-mir) (Cat.# A25576, Applied Biosystems). The 20 μL ddPCR reaction mixture

contained 10 μL of Bio-Rad 2x ddPCR Supermix for probes, 2.0 μL of diluted cDNA, 1.0 μL of individual miRNA-specific primer-probe mix and reaction volume adjusted to 20 μL by nuclease free water. Droplet digital PCR was performed using Bio-Rad Automated QX200 droplet digital PCR system as previously described [33].

## Statistical analysis

Statistical analysis was performed using GraphPad Quick Calcs t test calculator online software (http://www.graphpad.com/quickcalcs/ttest1.cfm). Analysis was performed using paired, two-tailed Student's t-test. P-values less than 0.05 were considered statistically significant.

## Results

### MiRNA levels in different fractions of human blood

We investigated miRNA concentrations in different components of blood. Blood was separated into 5 distinct fractions, RBCs, WBCs, platelets, PRP, and PFP. Subsequently, we extracted total RNA from each fraction and performed miRNA quantification using ddPCR. The total copy number of individual miRNAs in whole blood was calculated as the sum of all copies in all 5 fractions of blood (RBCs, WBCs, platelets, PRP, and PFP). Copy number in each fraction was divided by the total copy number and proportions reported as a percentage of the total miRNA copies. As depicted in Table 1, our results reveal that 99.9% of miRNA-451a is located within the RBCs. WBCs contain the second-highest amount (0.055%). In contrast, platelets, PRP, and PFP exhibit much lower miRNA-451a levels, accounting for 0.012%, 0.007%, and 0.0028%, respectively. The highest amount of miRNA-423-5p is also localized in RBCs (97.96%). However, only a 20% of miRNA-199a-3p is found in RBCs. Highest amount of miRNA-199a-3p (58.3%) is present in platelet fraction.

### Effect of hemolysis on plasma miRNA levels

After observing that over 97% of miRNA-451a and miRNA-423-5p is located in red blood cells, we investigated how hemolysis might influence the amount of these three miRNAs in PFP. The miRNA levels in hemolyzed versus non-hemolyzed PFP was compared using ddPCR. There was a statistically significant increase in miRNA-451a and miRNA-423-5p concentrations in hemolyzed samples as compared to non-hemolyzed samples (Fig 1). This indicates that hemolysis during sample storage or preparation, may significantly increase levels of miRNA-451a and miRNA-423-5p in PFP. However, miRNA-199a-3p levels in PFP remained unaffected after hemolysis (Fig 1).

### Effect of storage of whole blood and PRP at RT or 4°C on circulating miRNA-451a stability

To evaluate the impact of storage conditions on circulating miRNA-451a stability in both whole blood and PRP, we examined the levels of miRNA-451a at room temperature (RT) and

**Table 1. MiRNA concentrations in different fractions of human blood.**

| Blood fraction | miRNA-451a | | miRNA-423-5p | | miRNA-199a-3p | |
|---|---|---|---|---|---|---|
| | copies/100 μL blood | % in blood | copies/100 μL blood | % in blood | copies/100 μL blood | % in blood |
| Red Blood cells | $4.5 \times 10^{14}$ | 99.91% | $9.2 \times 10^{11}$ | 97.96% | $1.06 \times 10^{10}$ | 20% |
| White Blood Cells | $2.5 \times 10^{11}$ | 0.055% | $3.3 \times 10^9$ | 0.34% | $5 \times 10^9$ | 9.4% |
| Platelets | $5.7 \times 10^{10}$ | 0.012% | $1.1 \times 10^{10}$ | 1.2% | $3.09 \times 10^{10}$ | 58.3% |
| PRP | $3 \times 10^{10}$ | 0.007% | $0.5 \times 10^{10}$ | 0.5% | $6.47 \times 10^9$ | 12.2% |
| PFP | $1.3 \times 10^{10}$ | 0.0028% | $7.8 \times 10^7$ | 0.008% | $3 \times 10^7$ | 0.06% |

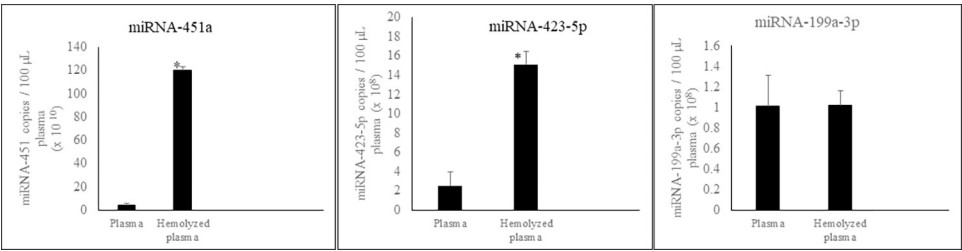

**Fig 1. Effect of blood hemolysis on circulating miRNA concentration in plasma.** Hemolyzed and non-hemolyzed PFP were prepared from each donor, plasma RNA extracted, and miRNA-451a, miRNA-423-5p and miRNA-199a-3p concentrations determined using respective ddPCR assays as described in Methods and Material section. n = 4; Error bars indicates SD; ***p = 0.0001.

4°C over a period of seven days. When whole blood was stored at RT (Fig 2A) or 4°C (Fig 2B), miRNA-451a levels were stable up to 3 days. At day 7, samples stored at RT and 4C showed 5.7- fold and 2.5-fold statistically significant increase in miRNA-451a levels, respectively. PRP stored at RT (Fig 2C) or 4C (Fig 2D) showed a time dependent statistically significant decrease in miRNA-451a levels, highlighting the inherent instability of this miRNA.

## Effect of storage of whole blood and PRP at RT and 4°C on circulating miRNA-423-5p concentration

We conducted an analysis of miRNA-423-5p levels in whole blood and PRP stored at RT and 4C. In whole blood samples stored at RT (Fig 3A), a significant increase in miRNA-423-5p concentration was observed at all time-points, compared to day-0 measurements. Whole

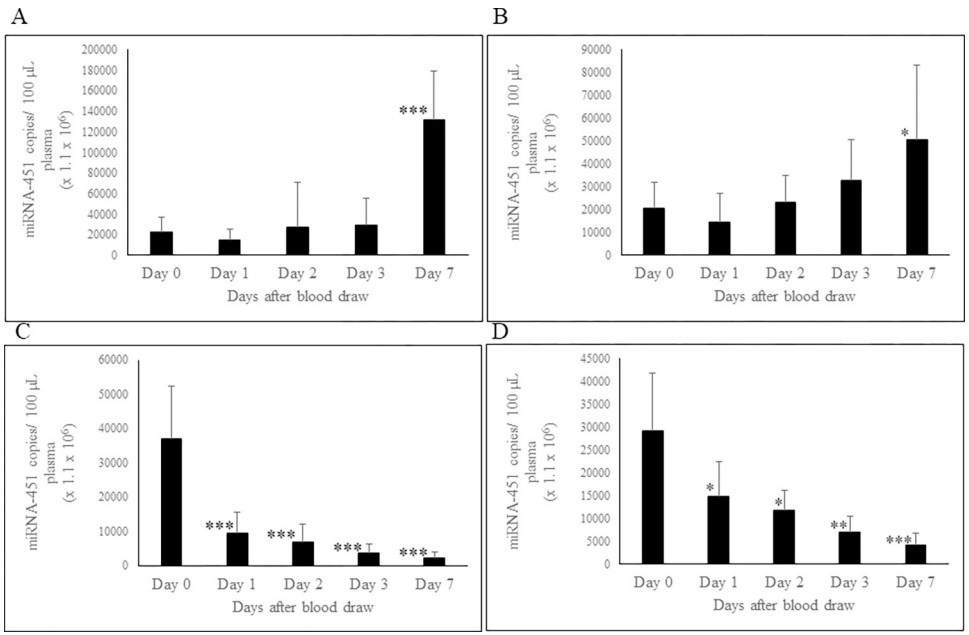

**Fig 2. Effect of storage of whole blood and PRP at RT (23°C) and at 4°C on stability of circulating miRNAs.** The miRNA-451a concentration in whole blood stored at RT (A) and 4C (B); miRNA-451a concentration in PRP stored at RT (C) and 4C (D). Blood was removed at indicated time points and plasma separated as described in "Methods and material section. Platelet rich plasma also removed at indicated time points and centrifuged at 16000 g for 15 minutes at RT to remove platelets, apoptotic bodies, and other cell debris. RNA was extracted as described and miRNA-451a concentration determined using ddPCR technology. n = 17. Error bars indicate SD. *p ≤ 0.04; **p ≤ 0.001; ***p ≤ 0.0001.

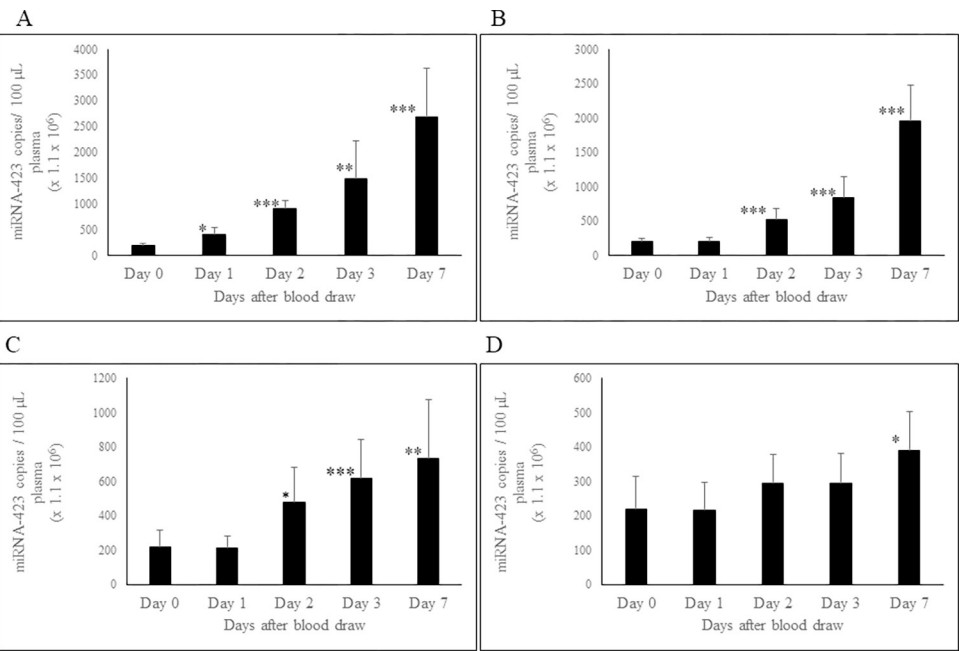

**Fig 3. Effect of storage of whole blood and PRP at RT and at 4˚C on stability of circulating plasma miRNA-423-5p.** The miRNA-423-5p concentration in whole blood stored at RT (A) and at 4˚C (B); miRNA-423-5p concentration in PRP stored at RT (C) and 4˚C (D). Blood and plasma samples were processed, and RNA extracted as described in Methods and material section. miRNA-423-5p concentration was determined using ddPCR technology. n = 8. Error bars indicates SD. *$p \leq 0.002$; **$p = 0.0003$; ***$p \leq 0.0001$.

blood stored at 4C (Fig 3B) showed no change up to day-1, but there was a statistically significant increase post day-2.

For PRP stored at RT (Fig 3C), there was no notable change in levels of miRNA-423-5p after day-1. However, concentrations at RT significantly increased at day-2, with consistent increases over days 3 and 7. PRP samples at 4˚C (Fig 3D) showed no change up to day-3, but a statistically significant increase was observed at day-7.

## Effect of storage of whole blood and PRP at RT and 4˚C on circulating miRNA-199a-3p concentration

We studied the effect of temperature and storage time on miRNA-199a-3p levels in whole blood and PRP. Whole blood stored at RT (Fig 4A) showed statistically significant increases in miRNA-199a-3p levels from day-2 through day-7. However, whole blood stored at 4˚C (Fig 4B) showed no change up to day-2, but a significant increase at days 3 and 7. PRP stored at RT (Fig 4C) exhibited statistically significant increases in miRNA-199a levels at days 2, 3, and 7, while this increase for PRP at 4C (Fig 4D) was apparent only at days 3 and 7.

## Effect of short-term storage of whole blood and PFP at RT on circulating miRNA-451a

As storage of whole blood and PRP at RT over the course of 7 days had an influence on miRNA expression levels, we then studied this impact by narrowing the scope of time to smaller increments. This is where a post-blood draw provides an optimal window in which samples could be processed early to attain consistent results. To this end, a measure of the stability of miRNA-451a, miRNA-423-5p and miRNA-199a-3p, in whole blood and PFP was

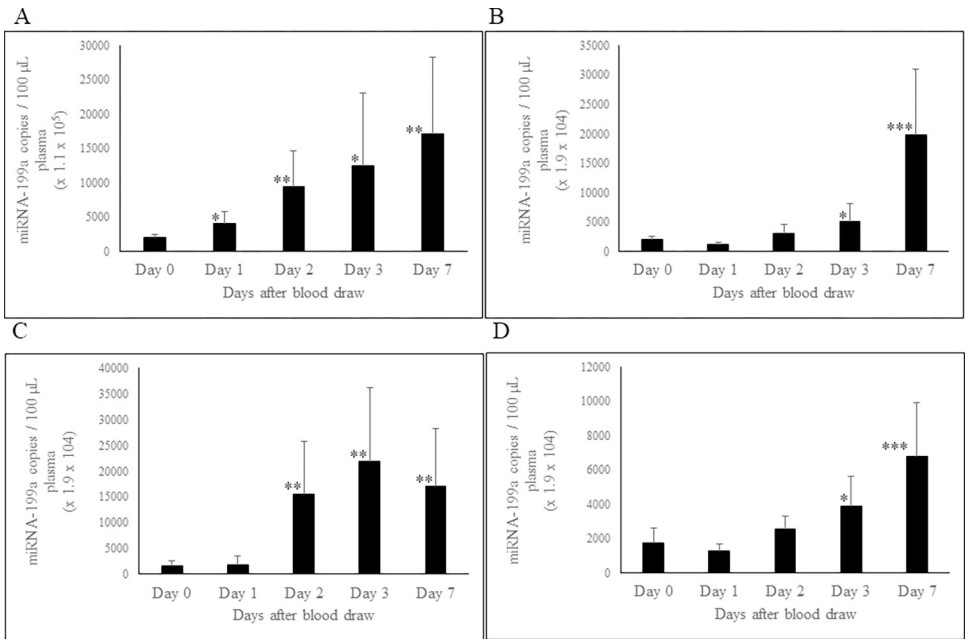

**Fig 4. Effect of storage of whole blood and PRP at RT and at 4°C on stability of circulating plasma miRNA-199a-3p.** The miRNA-199a-3p concentration in whole blood stored at RT (A) and at 4°C (B); miRNA-199a-3p concentration in PRP stored at RT (C) and at 4°C (D). Blood and plasma samples were processed, and RNA extracted as described in Methods and material section. miRNA-199a-3p concentration was determined using ddPCR technology. n = 8. Error bars indicates SD. $^*p \leq 0.02$; $^{**}p \leq 0.008$; $^{***}p \leq 0.0006$.

assessed at RT storage conditions over shorter time intervals at 2, 6, 12, 24, 36, 48, and 72 hours. MiRNA-451a concentrations remained relatively stable in whole blood at RT for up to 48 hours, and at 72 hours there was a statistically significant increase compared to the 2-hour initial level (Fig 5A). MiRNA-451a level in PFP showed a nonsignificant decrease at 6 hours. However, starting at 12 hours, we observed a statistically significant constant decline in plasma levels of miRNA-451a (Fig 5B).

## Effect of short-term storage of whole blood and PFP at RT on circulating miRNA-423-5p and miRNA-199a-3p

MiRNA-423-5p level in whole blood was stable initially up to 24 hours, thereafter, showing a statistically significant increase over time (Fig 6A). MiRNA-423-5p level in PFP was stable up

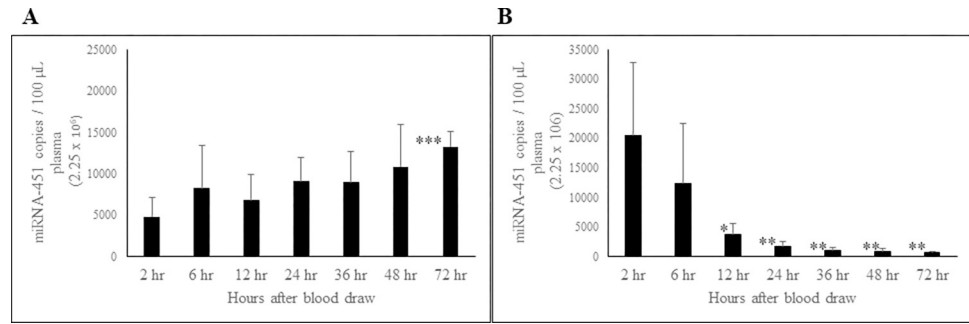

**Fig 5. Effect of short-term storage of whole blood and PFP at RT on circulating miRNA-451a levels.** The miRNA-451a concentration in whole blood (A) and PFP (B) samples stored at RT. n = 6; Error bars indicate SD; $^*p \leq 0.04$; $^{**}p \leq 0.02$; $^{***}p \leq 0.0015$.

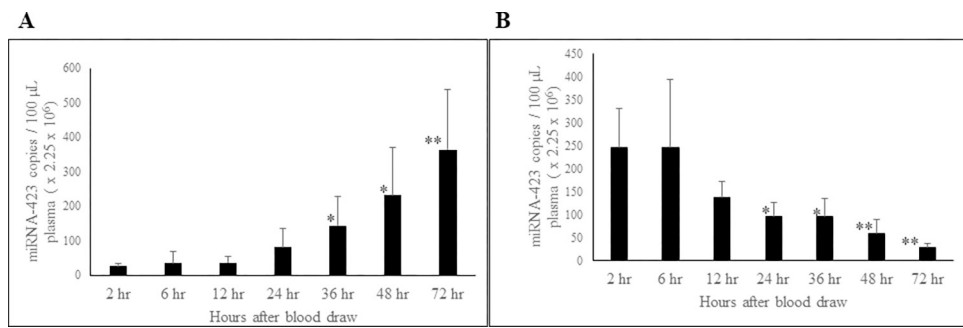

**Fig 6. Effect of short-term storage of whole blood and PFP at RT on circulating miRNA-423-5p concentration.**
The miRNA-423-5p concentration compared for whole blood (A) and PFP (B) samples stored at RT. n = 6; Error bars indicates SD; *p < 0.04; **p = 0.009.

to 6 hours, with a non-significant decrease in concentration at 12 hours. However, there was a statistically significant decline in miRNA-423-5p levels starting at 24 hours (Fig 6B).

In whole blood miRNA-199a-3p level was stable up to 24 hours and thereafter showed a statistically significant increase (Fig 7A). However, levels of miRNA-199a-3p in PFP was stable throughout the 72-hour period (Fig 7B).

## Effect of -20˚C and -80˚C storage conditions on PFP miRNA-451a levels

Given the observed instability of miRNA-451a in plasma both at RT and 4˚C, we conducted a stability study at lower temperatures, specifically -20˚C and -80˚C, over a period of 7 days. As depicted in Fig 8A, PFP storage at -20C showed a statistically significant decrease in miRNA-451a levels at day 2 and day 7, indicating highly unstable nature of this miRNA. However, miRNA-451a concentration in PFP stored at -80˚C was stable during the 7-day experimental period (Fig 8B).

Since miRNA-423-5p in PFP was unstable at RT, we investigated the stability of miRNA-423-5p and miRNA-199a-3p and observed both miRNAs to be stable in PFP samples stored at -20˚C or -80˚C for up to 7 days, as applicable in case of miRNA-451a (data not shown).

## Discussion

Circulating miRNA-451a exhibits significant potential as a biomarker for diagnosing various diseases, including cancer, chronic kidney disease, and autism spectrum disorder [12–17].

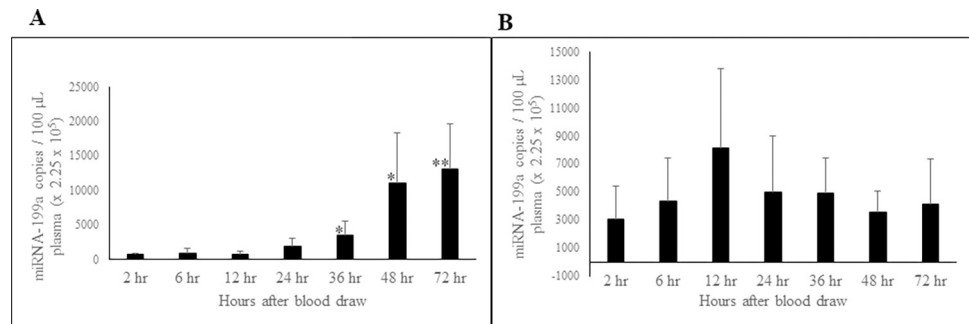

**Fig 7. Effect of short-term storage of whole blood and PFP at RT on circulating miRNA-199a-3p concentration.**
The miRNA-199a-3p concentration compared for whole blood (A) and PFP (B) samples stored at RT. n = 6; Error bars indicates SD; *p < 0.04; **p = 0.009.

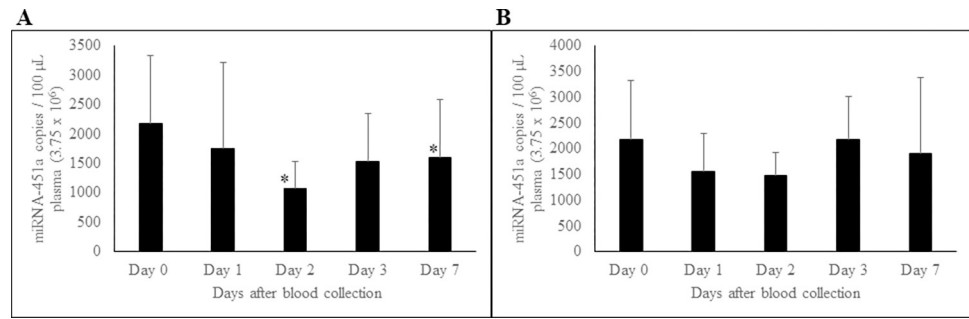

**Fig 8. Effect of storage of PFP at -20˚C and -80˚C on circulating miRNA-451a.** The miRNA-451a levels in PFP stored at -20˚C (A) and at -80˚C (B), monitored for up to 7 days. n = 6; Error bars indicate SD. *p < 0.02.

However, one of the major challenges in utilizing circulating miRNAs for developing novel diagnostic assays is preanalytical variability. This study addresses several preanalytical factors that can impact assay accuracy, such as blood sample hemolysis, the way plasma is separated from blood, sample storage time, and temperature. We investigated the distribution of miRNA-451a, miRNA-423-5p and miRNA-199a-3p in different blood fractions, including RBCs, WBCs, platelets, PRP, and PFP. Based on our results we devised a set of empirically driven suggestions for sample processing to avoid variations in these circulating miRNA concentrations. It is imperative that preanalytical factors impacting diagnostic assay results might vary across different miRNAs with respect to their cellular origin or circulation in cell free plasma.

Our findings reveal that an overwhelming 99.9% of miRNA-451a and over 97.6% of miRNA-423-5p is localized within RBCs, confirming earlier work [34]. In contrast, plasma contains only minute amounts of these miRNAs compared to RBCs. Consequently, our results demonstrated that hemolysis of the blood sample, a preanalytical factor, exhibited a statistically significant increase in levels of these miRNAs as compared to non-hemolyzed plasma, highlighting the unacceptability of using hemolyzed samples in miRNA-451a or miRNA-423-5p based assays. However, hemolysis had no effect on miRNA-199a-3p concentration in PFP. All three-miRNA concentration in platelets are several fold higher compared to PFP. Therefore, we recommend that high quality (second spin) platelet free plasma be used for miRNA detection assays. Also, when it is necessary to ship the blood sample to a different location for diagnostic testing, platelet free plasma (PFP) be preferred instead of whole blood since mechanical breakdown of RBCs, WBCs or platelets during transit may cause a drastic increase in miRNA levels, thereby obfuscating assay results.

Furthermore, our investigation delves into the effects of storage time and temperature on miRNA-451a levels in blood and PRP. When PRP was stored at room temperature (RT) and 4˚C for up to 7 days, there was a consistent statistically significant decrease in miRNA-451a levels, indicating the inherent instability of this miRNA at RT or 4C. Therefore, it is not advisable to store blood or plasma at RT or 4C. Our short-term study shows that storing PFP at RT for 6 hours caused a drop in miRNA-451a level though it was not statistically significant. However, at 12 hours and thereafter up to 72 hours there was a steady decline in concentration. Hence it is recommended that blood be processed within 6 hours of blood draw.

The question of why miRNA-451a levels remain stable in stored whole blood but decline in platelet rich plasma (PRP) can potentially be explained by the fact that RBCs are capable of releasing exosomes and other extracellular vesicles that contain miRNA-451a into circulation, as previously reported [35]. Thus, any decrease in miRNA-451a levels that is due to its

instability/degradation may be compensated for by what is released from RBCs or RBCs undergoing hemolysis post-blood draw.

In addition to miRNA-451a, we also investigated the stability of two other clinically important miRNAs that are abundant in plasma: miRNA-199a-3p and miRNA-423-5p. When whole blood was stored at RT and 4˚C, the levels of both miRNAs increased over time. This increase is likely due to release of extracellular vesicles from blood cells, which carry a diverse population of miRNAs. This behavior is different when compared to miRNA-451a that showed an increase only at day-7.

The increase in the levels of miRNA-423-5p and miRNA-199a-3p in PRP can be attributed to the fact that both miRNAs are among top 20 most abundant miRNAs in platelets [36]. Consequently, the observed increase may result from the release of exosomes from platelets during storage. Additionally, dead cells, apoptotic bodies, and other cellular debris may also release miRNAs into plasma during storage. Additionally, dead cells, apoptotic bodies and other cellular debris may also release miRNAs in plasma during storage. Therefore, we recommend using high quality PFP for measuring circulating cell-free miRNA, achieved through high-speed centrifugation (at 16,000 g for 10 minutes) to remove most platelets, apoptotic bodies, dead cells, and other cellular debris.

Our study was designed to examine the effects of storage at days 1, 2, 3, and 7. As miRNA-451a concentration in plasma showed a significant drop around day-1, in another study we investigated short-term storage effects on whole blood and PFP stored at RT. In whole blood, miRNA-451a level overall remains stable up to 72 hours, showing a trend toward a significant increase. However, in PFP, a decline in miRNA-451a levels is observed becoming significant at the 12-hour mark. These findings underscore the importance of promptly processing plasma samples for miRNA-451a detection.

Since miRNA-451a was unstable at RT and 4˚C, we explored the stability of platelet-free plasma at -20˚C and -80˚C over a 7-day period. Our data shows that miRNA-451a concentration decreased at -20˚C, specially at day 2 and 7, indicating that at -20˚C, PFP miRNA-451a was not stable. However, PFP stored at -80˚C was stable throughout the 7-day period indicating that miRNA-451a in PFP is stable at -80˚C at least for 7 days.

In this study, using a set of three important miRNAs, we demonstrated that miRNAs could behave differentially with respect to the impact of preanalytical factors. Therefore, it is important to investigate the effect of preanalytical factors on assay accuracy on all clinically important miRNAs. Our future studies will be directed towards utilizing miRNA-microarray and miRNA-sequencing technologies to further investigate the preanalytical factors that may have impact on accurate quantification of clinically important miRNAs.

## Conclusions

In conclusion, our study underscores the need for precautions when using miRNA-451a, miRNA-423-5p and miRNA-199a-3p as biomarkers in diagnostic assays. A common perception is that miRNAs are quite stable especially due to secondary structure and interactions with their targeting complex [37], but here we observe that preanalytical factors such as hemolysis, storage time and temperature may change miRNA levels. Careful monitoring and control of the preanalytical phase are essential during blood collection and processing to prevent hemolysis, which can release miRNA into plasma and compromise assay validity. To eliminate platelets and other cell debris, the recommended approach is the two-step centrifugation protocol as previously described [32]. Additionally, we emphasize the significance of immediate processing of blood samples to obtain plasma and extract RNA. When plasma storage is necessary, particularly for batch processing, it is advisable to store PFP at -80˚C. Importantly, blood

should never be stored at freezing temperatures, as freezing and thawing can cause hemolysis leading to increased miRNA concentrations in plasma. To maintain the integrity of plasma samples, it is recommended that the latter are devoid of platelets (PFP), as platelet-contents released during freezing and thawing may also lead to increase in miRNA levels.

## Future directions

Although miRNAs are considered stable entities specially when packaged and transported via exosomes, their expression may vary depending on their origin or disease specific miRNA types. Therefore, further research must evaluate differential expression of clinically important miRNAs, and how they influence other miRNA biomarkers in proximity, in exosomes circulating in human plasma derived from patients and healthy controls. Further insights on how miRNA's intrinsic stability may be impacted due to differential regulation at the level of genesis or preprocessing events, will help improvise future miRNA-biomarker based disease diagnostics.

## Acknowledgments

We wish to thank Dr. Dominic Cosgrove, Center for Sensory Neuroscience, Boys Town National Research Hospital, Omaha NE USA for his advice and help during this project. We gratefully acknowledge the assistance from Rebecca Cash for IRB application. We want to thank Dan Meehan, Center for Sensory Neuroscience for his help.

## Author Contributions

**Conceptualization:** M. Rohan Fernando.

**Data curation:** Dinesh S. Chandel.

**Formal analysis:** Dinesh S. Chandel, Wesley A. Tom, Chao Jiang, Gary Krzyzanowski, Appolinaire Olou.

**Funding acquisition:** M. Rohan Fernando.

**Investigation:** Dinesh S. Chandel, Chao Jiang, Gary Krzyzanowski, Nirmalee Fernando.

**Methodology:** Dinesh S. Chandel, Wesley A. Tom, Chao Jiang, Gary Krzyzanowski, Nirmalee Fernando, Appolinaire Olou.

**Software:** Wesley A. Tom.

**Supervision:** M. Rohan Fernando.

**Validation:** Wesley A. Tom.

**Visualization:** Wesley A. Tom, Appolinaire Olou.

**Writing – original draft:** M. Rohan Fernando.

**Writing – review & editing:** Dinesh S. Chandel, Chao Jiang, Gary Krzyzanowski, Nirmalee Fernando, Appolinaire Olou.

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
