## [Decision Letter · Decision Letter 0]

12 Mar 2024

PONE-D-24-05265Preanalytical considerations for clinical assays of circulating human miRNA-451a for diagnostic purposes.PLOS ONE

Dear Dr. Fernando, 

Thank you for submitting your manuscript to PLOS ONE. After careful consideration, we feel that it has merit but does not fully meet PLOS ONE’s publication criteria as it currently stands. Therefore, we invite you to submit a revised version of the manuscript that addresses the points raised during the review process.

We look forward to receiving your revised manuscript.

Kind regards,

Dr. Himanshu Gupta

Academic Editor

PLOS ONE

Journal Requirements:

https://www.preprints.org/subject/browse/medicine_and_pharmacology/clinical_medicine_130?

In your revision ensure you cite all your sources (including your own works), and quote or rephrase any duplicated text outside the methods section. Further consideration is dependent on these concerns being addressed.

"This research was funded by a research grant from Ryan Foundation given to MRF."

"We wish to thank Dr. Dominic Cosgrove, Center for Sensory Neuroscience, Boys Town National

Research Hospital, Omaha NE USA for his advice and help during this project. We gratefully

acknowledge the assistance from Rebecca Cash for IRB application. We want to thank Dan

Meehan, Center for Sensory Neuroscience for his help. This research was funded by a research

grant from the Ryan Foundation to MRF."

"This research was funded by a research grant from Ryan Foundation given to MRF."

8. We notice that your supplementary figures are uploaded with the file type 'Figure'. Please amend the file type to 'Supporting Information'. Please ensure that each Supporting Information file has a legend listed in the manuscript after the references list.

Reviewers' comments:

Reviewer's Responses to Questions

**Comments to the Author**

1. Is the manuscript technically sound, and do the data support the conclusions?

Reviewer #1: Yes

Reviewer #2: Yes

2. Has the statistical analysis been performed appropriately and rigorously? 

Reviewer #1: Yes

Reviewer #2: Yes

3. Have the authors made all data underlying the findings in their manuscript fully available?

Reviewer #1: Yes

Reviewer #2: Yes

4. Is the manuscript presented in an intelligible fashion and written in standard English?

Reviewer #1: Yes

Reviewer #2: Yes

5. Review Comments to the Author

Reviewer #1: The manuscript is related to preanalytical factors that can affect circulating miRNAs, an important issue to be considered, especially due to higher variability in the assays.

In order to publish this article, some corrections must be made:

Objectives

The authors should clarify the objectives, as the miRNA-451a was considered the main miRNA of this work (considering the title), but all over the text and even in the last phrase of introduction other miRNAs are mentioned. Are the results of other miRNAs (miRNA-423-5p and miRNA-199a-3p) relevant to the manuscript? If so they should be addressed in the objectives and introduction.

Introduction

Information related to miRNA-423-5p and miRNA-199a-3p should be included in the introduction instead of results.

Methods and Materials

The number (n) of blood samples used to extract miRNA from whole blood, PRP, PFP should be listed at materials and methods. It is not clear all over the text. Was this “n” enough to support the results?

Results

The information from “MiRNA-423-5p is known for its high abundance in plasma and has been reported as an endogenous control for the quantification of circulating miRNAs in certain cancers” and “MiRNA-199a-3p is another important miRNA that is highly abundant in blood, shown as a promising diagnostic and prognostic marker in glioma” should be moved to introduction.

Are there any result related to effect of hemolysis on miRNA-423-5p and miRNA-199a-3p?

In case the miRNA-423-5p and miRNA-199a-3p will be included in the objectives, the supplemental figures should be moved to results, maybe including different bar colors to the previous ones.

Conclusion

It is important to include a conclusion related to miRNA-423-5p and miRNA-199a-3p results or that the preanalytical interferences may vary according to miRNA used.

Reviewer #2: Report on "Preanalytical Considerations for Clinical Assays of Circulating Human miRNA-451a for Diagnostic Purposes"

The manuscript investigates preanalytical factors affecting the accurate quantification of miRNA-451a in human blood samples, crucial for its diagnostic potential. Key findings include the predominant localization of miRNA-451a in red blood cells (RBCs) and significant hemolysis impact on plasma concentration. Storage conditions also play a role, with RT and 4ºC affecting platelet-rich plasma (PRP) and platelet-free plasma (PFP) stability, while -80ºC maintains stability up to 7 days.

Review:

Sample Size Clarity: The manuscript utilizes normal healthy samples; however, the number of samples is not specified. Clear reporting of the sample size is essential for result interpretation and study validity.

Enhanced Study Design: To strengthen the study's impact, it is suggested to consider incorporating different patient or disease samples for comparison with controls. This addition would provide valuable insights into the specificity and diagnostic potential of miRNA-451a across various health conditions.

Conclusion:

The study offers valuable insights into preanalytical considerations for miRNA-451a quantification, emphasizing the need for careful handling to ensure accurate results. Addressing the highlighted points could further enhance the robustness and applicability of the findings.

6. PLOS authors have the option to publish the peer review history of their article (what does this mean?). If published, this will include your full peer review and any attached files.

Reviewer #1: No

Reviewer #2: No

---

## [Author Response · Author response to Decision Letter 0]

11 Apr 2024

Response to Reviewers

RE: Manuscript # PONE-D-24-05265, titled “Preanalytical considerations for clinical assays of circulating human miRNA-451a for diagnostic purposes”.

April 10, 2024.

Professor Himanshu Gupta

Academic Editor

PLOS ONE

Dear Professor Gupta,

We would like to take this opportunity to thank you for considering our manuscript for peer review, and the opportunity for a revised submission. Your valuable editorial comments have helped us improve our article as per PLOS ONE format. We welcome all suggestions and comments made by both the reviewers as equally important and have tried to address them all with incorporated necessary changes in our revised submission. Below is the pointwise response in brief, to all queries/comments received in your review report.

Editors’ comments and reply to editors’ comments.

Rebuttal letter enclosed.

Uploaded the track-change version of the revised manuscript.

Final revised version without tracked changes, uploaded.

Journal Requirements: When submitting your revision, we need you to address these additional requirements.

• The manuscript has been formatted following PLOS One requirements.

https://www.preprints.org/subject/browse/medicine_and_pharmacology/clinical_medicine_130?

In your revision ensure you cite all your sources (including your own works), and quote or rephrase any duplicated text outside the methods section. Further consideration is dependent on these concerns being addressed.

• The overlapping text is a preprint (unpublished) generated from a past submission to a different journal, as some journals encourage authors to upload a preprint during submissions. However, our manuscript was rejected by that journal and for this current PLOS ONE submission, we conducted additional experiments and updated extensively, comparing all three miRNAs for this study. This is also reflected within the revised title. As suggested, our current revised submission to PLOS ONE has citations updated and duplicated texts have been either omitted or rephrased. 

• The funding information has been checked and grant number updated.

"This research was funded by a research grant from Ryan Foundation given to MRF."

• The revised Financial Disclosures statement has been added separately. “This research was funded by a research grant (RFLG-2019-2023-0003) from the Ryan Foundation to MRF. The funders had no role in study design, data collection and analysis, decision to publish, or preparation of the manuscript.”

"We wish to thank Dr. Dominic Cosgrove, Center for Sensory Neuroscience, Boys Town National

Research Hospital, Omaha NE USA for his advice and help during this project. We gratefully

acknowledge the assistance from Rebecca Cash for IRB application. We want to thank Dan

Meehan, Center for Sensory Neuroscience for his help. This research was funded by a research

grant from the Ryan Foundation to MRF."

 "This research was funded by a research grant from Ryan Foundation given to MRF."

• The funding information has now been sated under the assigned section separate from acknowledgements.

• The ORCID iD has been linked and validated in Editorial manager.

• All captions/supporting figures (now included in results) have been shifted to the main text.

1. 8. We notice that your supplementary figures are uploaded with the file type 'Figure'. Please amend the file type to 'Supporting Information'. Please ensure that each Supporting Information file has a legend listed in the manuscript after the references list.

• Considering valuable suggestions received from both reviewers, we now have shifted all of previous supplement results/figures describing the other two miRNAs to the main manuscript-text, and relabeled them in the order cited. 

Response to reviewer 1

Reviewer #1: The manuscript is related to preanalytical factors that can affect circulating miRNAs, an important issue to be considered, especially due to higher variability in the assays.

• Thank you for an extensive review of our submission to PLOS ONE, recognizing the importance of this study in context of miRNA stability during preanalytical phase, pertaining to high variability observed in clinical assays. 

In order to publish this article, some corrections must be made:

Objectives

1. The authors should clarify the objectives, as the miRNA-451a was considered the main miRNA of this work (considering the title), but all over the text and even in the last phrase of introduction other miRNAs are mentioned. Are the results of other miRNAs (miRNA-423-5p and miRNA-199a-3p) relevant to the manuscript? If so they should be addressed in the objectives and introduction.

• Thank you for pointing out this weakness in our manuscript. By changing the title, objectives in the abstract and the introduction we tried to address this issue. The title, objectives in the abstract and the introduction was changed to include all three miRNAs. Considering the other two miRNAs equally important, we have updated the manuscript placing these miRNAs along with the miRNA-451a in all sections. 

2. Introduction

Information related to miRNA-423-5p and miRNA-199a-3p should be included in the introduction instead of results.

• As per reviewers’ suggestion, the other two miRNAs (miR-423/199a) now are also highlighted in the Intro/Discussion sections, with a revised results section. 

3. Methods and Materials

The number (n) of blood samples used to extract miRNA from whole blood, PRP, PFP should be listed at materials and methods. It is not clear all over the text. Was this “n” enough to support the results?

• As per your suggestion Methods and materials section (Blood samples section) was modified to include number of blood donors used for each experiment.

4. Results

The information from “MiRNA-423-5p is known for its high abundance in plasma and has been reported as an endogenous control for the quantification of circulating miRNAs in certain cancers” and “MiRNA-199a-3p is another important miRNA that is highly abundant in blood, shown as a promising diagnostic and prognostic marker in glioma” should be moved to introduction.

• As per your suggestion we moved that information from results section to introduction. 

Are there any result related to effect of hemolysis on miRNA-423-5p and miRNA-199a-3p?

• Yes, we have modified Fig 1 to include the results for miRNA-423-5p and miRNA-199a-3p.

In case the miRNA-423-5p and miRNA-199a-3p will be included in the objectives, the supplemental figures should be moved to results, maybe including different bar colors to the previous ones.

• As per your suggestion we moved all supplemental Figures to main text. 

Conclusion

It is important to include a conclusion related to miRNA-423-5p and miRNA-199a-3p results or that the preanalytical interferences may vary according to miRNA used.

• This section is now updated, with all three miRNAs compared, as preanalytical factors may vary across miRNA families. Since miRNAs stability can be inherently different due to their specific origin or preprocessing events during maturation and function, we emphasize that levels of disease specific/individual miRNAs should therefore be interpreted with caution. 

Reviewer #2: 

Report on "Preanalytical Considerations for Clinical Assays of Circulating Human miRNA-451a for Diagnostic Purposes"

The manuscript investigates preanalytical factors affecting the accurate quantification of miRNA-451a in human blood samples, crucial for its diagnostic potential. Key findings include the predominant localization of miRNA-451a in red blood cells (RBCs) and significant hemolysis impact on plasma concentration. Storage conditions also play a role, with RT and 4ºC affecting platelet-rich plasma (PRP) and platelet-free plasma (PFP) stability, while -80ºC maintains stability up to 7 days.

• Thank you for an extensive review of our submission to PLOS ONE, recognizing the importance of this study in context of miRNA stability during preanalytical phase, pertaining to high variability observed in clinical assays. 

Review:

Sample Size Clarity: The manuscript utilizes normal healthy samples; however, the number of samples is not specified. Clear reporting of the sample size is essential for result interpretation and study validity.

As per your suggestion Methods and materials section (Blood samples section) was modified to include number of blood donors used for each experiment.

 Enhanced Study Design: To strengthen the study's impact, it is suggested to consider incorporating different patient or disease samples for comparison with controls. This addition would provide valuable insights into the specificity and diagnostic potential of miRNA-451a across various health conditions.

• We understand extending our analysis on patient samples may enhance study’s impact. This remains our future goal with ongoing biomarker research. Our current IRB approval is only for normal blood donors and not for patient samples. We hope to continue this project with patient samples in near future with a new IRB protocol.

Conclusion: The study offers valuable insights into preanalytical considerations for miRNA-451a quantification, emphasizing the need for careful handling to ensure accurate results. Addressing the highlighted points could further enhance the robustness and applicability of the findings. 

• Thank you for your valuable comments. We hope that the revisions we made to this version of our manuscript enhanced the applicability of our findings in laboratory medicine. 

---

## [Decision Letter · Decision Letter 1]

29 Apr 2024

Preanalytical considerations for clinical assays of circulating human miRNA-451a, miRNA-423-5p and miRNA-199a-3p for diagnostic purposes.

PONE-D-24-05265R1

Dear Dr. Fernando,

We’re pleased to inform you that your manuscript has been judged scientifically suitable for publication and will be formally accepted for publication once it meets all outstanding technical requirements.

Kind regards,

Himanshu Gupta, Ph.D., MRSB, MASTMH

Assistant Professor, Department of Biotechnology,

GLA University, Mathura, India.

Academic Editor

PLOS ONE

Additional Editor Comments (optional):

Reviewers' comments:

Reviewer's Responses to Questions

**Comments to the Author**

1. If the authors have adequately addressed your comments raised in a previous round of review and you feel that this manuscript is now acceptable for publication, you may indicate that here to bypass the “Comments to the Author” section, enter your conflict of interest statement in the “Confidential to Editor” section, and submit your "Accept" recommendation.

Reviewer #1: All comments have been addressed

Reviewer #2: All comments have been addressed

2. Is the manuscript technically sound, and do the data support the conclusions?

Reviewer #1: Yes

Reviewer #2: Yes

3. Has the statistical analysis been performed appropriately and rigorously? 

Reviewer #1: Yes

Reviewer #2: Yes

4. Have the authors made all data underlying the findings in their manuscript fully available?

Reviewer #1: Yes

Reviewer #2: Yes

5. Is the manuscript presented in an intelligible fashion and written in standard English?

Reviewer #1: Yes

Reviewer #2: Yes

6. Review Comments to the Author

Reviewer #1: The authors corrected the manuscript accordingly to our suggestions.

The manuscript is suitable to publication at Plos One.

Reviewer #2: Thank you for informing me. The modifications made to the paper will undoubtedly enhance its quality and relevance. Looking forward to its publication and contributing to the field of preanalytical considerations for clinical assays of circulating human miRNA.

7. PLOS authors have the option to publish the peer review history of their article (what does this mean?). If published, this will include your full peer review and any attached files.

Reviewer #1: No

Reviewer #2: No

---

## [Editor Report · Acceptance letter]

2 May 2024

PONE-D-24-05265R1 

PLOS ONE

Dear Dr. Fernando, 

I'm pleased to inform you that your manuscript has been deemed suitable for publication in PLOS ONE. Congratulations! Your manuscript is now being handed over to our production team.

Kind regards, 

on behalf of

Dr. Himanshu Gupta 

Academic Editor

PLOS ONE